# The Efficacy of Re-Warm-Up Practices during Half-Time: A Systematic Review

**DOI:** 10.3390/medicina57090976

**Published:** 2021-09-17

**Authors:** Daniel González-Devesa, Alejandro Vaquera, David Suárez-Iglesias, Carlos Ayán-Pérez

**Affiliations:** 1Faculty of Physical Activity and Sports Sciences, University of León, 24007 León, Spain; danidevesa4@gmail.com; 2VALFIS Research Group, Institute of Biomedicine (IBIOMED), University of León, 24007 León, Spain; avaquj@unileon.es; 3School of Sport & Exercise Science, University of Worcester, Worcester WR2 6AJ, UK; 4Departamento de Didácticas Especiais, Universidade de Vigo, Well-Move Research Group, Galicia Sur Health Research Institute (IIS Galicia Sur), SERGAS-UVIGO, 36310 Pontevedra, Spain; cayan@uvigo.es

**Keywords:** performance, active recovery, rest period

## Abstract

*Background and Objectives*: The passive nature of rest breaks in sport could reduce athletes’ performance and even increase their risk of injury. Re-warm-up activities could help avoid these problems, but there is a lack of research on their efficacy. This systematic review aimed at analyzing the results of those randomized controlled trials (RCTs) that provided information on the effects of re-warm-up strategies. *Materials and Methods*: Four electronic databases (Web of Science, Scopus, PubMed, and SPORTDiscus) were searched from their inception to January 2021, for RCTs on the effects of re-warm-up activities on sports performance. Interventions had to be implemented just after an exercise period or sports competition. Studies that proposed activities that were difficult to replicate in the sport context or performed in a hot environment were excluded. Data were synthesized following PRISMA guidelines, while the risk of bias was assessed following the recommendations of the Cochrane Collaboration. *Results*: A total of 14 studies (178 participants) reporting data on acute or short-term effects were analyzed. The main outcomes were grouped into four broad areas: physiological measures, conditional abilities, perceptual skills, and sport efficiency measures. The results obtained indicated that passive rest decreases physiological function in athletes, while re-warm-up activities could help to improve athletes’ conditional abilities and sporting efficiency, despite showing higher fatigue levels in comparison with passive rest. The re-warm-up exercise showed to be more effective than passive rest to improve match activities and passing ability. *Conclusions*: Performing re-warm-up activities is a valuable strategy to avoid reducing sports performance during prolonged breaks. However, given that the methodological quality of the studies was not high, these relationships need to be further explored in official or simulated competitions.

## 1. Introduction

Warm-up is a preparatory exercise period that helps athletes adapt to the intensity demanded by competition, improve sports performance, and decrease the risk of injury [1]. However, the structure of modern sports often reduces the efficacy of warming up due to the inactivity of the athletes during the competition breaks (also known as half times or quarters) [2]. Indeed, the typically passive nature of these periods can lead to decrements in sports performance [3] and even an increased risk of injury [4].

During the aforementioned breaks, it has been recommended to carry out brief re-warm-up activities (RW-U) aimed at attenuating the reduction in temperature and sports performance [5]. Nevertheless, performing RW-U activities does not seem to be a widely used performance-enhancing strategy since coaches and athletes tend to utilize this competitive pause to deliver tactical instructions, rehydrate and receive medical attention [6].

In addition, there is little information on how to perform an accurate RW-U routine due to the lack of scientific evidence available. Consequently, athletes and coaches continue to design their routines based solely on their experience [7]. This is a matter of concern since the passive nature of the competition breaks leads to a decline in physical and cognitive performance and increases the risk of injury [8]. Thus, coaches and sports scientists need to have valuable information at hand regarding how to develop RW-U to optimize athletes’ performance and reduce injury risk.

This goal can be accomplished by performing systematic reviews that collectively summarize all scientific evidence on the topic. To date, various reviews have emerged to present scientific evidence on the effects of including RW-U regimes during competition breaks and offer guidelines for its preparation. However, the findings provided were somehow limited. For instance, Hammami et al. [5] specifically reviewed investigations focused on soccer, while Silva et al. [7] examined the efficacy of RW-U activities in explosive sports and exclusively included studies involving experienced athletes, regardless of their design. The latter is a point to consider producing the highest level of scientific evidence, as systematic reviews should be based on the inclusion and detailed analysis of the randomized controlled trials (RCTs) published on the subject so far [8]. Finally, Russell et al. [9] carried out a broad review of the effects of different strategies to improve sports performance after breaks. However, since their review was narrative, it lacked a systematic search and a quality appraisal of the revised studies. Moreover, it mainly centered on team sports, and the most recent articles were published more than six years ago.

Given this situation, it seems necessary to update the scientific evidence regarding the efficacy of RW-U as a strategy for avoiding the expected decrement in sports performance that takes place after a resting period. Therefore, the purpose of this study was to review and critically analyze the results of those RCTs that provided information on the effects of RW-U on parameters related to sports performance, regardless of the sport modality investigated and the athletes’ level.

## 2. Materials and Methods

### 2.1. Design

This systematic review was conducted according to the PRISMA (Preferred Reporting Items for Systematic reviews and Meta-Analyses) guidelines [10]. This review was registered with the Open Science Framework (OSF), doi: 10.17605/OSF.IO/M248F.

### 2.2. Search Strategy

We identified studies published before January 2021 using four databases: Web of Science (Clarivate™), Scopus^®^ (Elsevier B.V.), PubMed (United States National Library of Medicine), and SPORTDiscus (EBSCO Industries Inc.). We employed the following search strategy and keywords: [“re-warm-up”] OR [“half-time strategy”] OR [“second-half”] AND [“warm-up”] OR [“RW-U strategy”]. A full description of input arguments used in each database is also provided (Electronic Appendix A).

### 2.3. Eligibility Criteria

Research articles were included or excluded using criteria defined with the PICO (Population, Intervention, Comparison and Outcome) criteria (Table 1), and the literature searches were limited to RCTs providing information about the effects of RW-U activities on sports performance when implemented just after an exercise period or sports competition. Theses, dissertations, and conference abstracts and proceedings were also excluded. There were no restrictions on written language, but studies were required to be written in English, Portuguese or Spanish abstract and published in a peer-reviewed journal.

### 2.4. Study Selection

Two authors screened the titles and abstracts of the identified records for eligibility. After independently reviewing the selected studies for inclusion, both authors compared them to reach an agreement. Once they reached an agreement, a full-text copy of every potentially relevant study was obtained. If it was unclear whether the study met the selection criteria, advice was sought from a third author to achieve consensus. Additionally, we manually screened the full texts of the studies that met the inclusion criteria and different systematic reviews for any additional relevant references.

### 2.5. Data Extraction

General details on the study title, authors, and design were extracted. Also, available data on the number of participants, age, sex, sport, category, height, weight, and information on the intervention, test or tests, and significant results of the studies were collected. For this purpose, the effects of RW-U activities were classified as “acute” (just after the break ended) or “short-term” (during the second period of the sports competition). All this information was extracted from the original reports by two researchers and tabulated into a matrix. A third investigator checked this process. 

### 2.6. Quality Appraisal

Two authors independently assessed the risk of bias of each study against key criteria: random sequence generation, allocation concealment, blinding of participants, personnel and outcomes, incomplete outcome data, selective outcome reporting, and other sources of bias, following methods recommended by the Cochrane collaboration [10]. The following classifications were used: low risk, high risk, or unclear risk (either lack of information or uncertainty regarding the potential for bias). The authors resolved disagreements by consensus, and they consulted a third author to help them if necessary. Review Manager software (RevMan, The Nordic Cochrane Centre, Copenhagen, Denmark) Version 5.4. was utilized to create risk-of-bias graphs (Figure 1 and Figure 2). 

## 3. Results

We obtained 693 records from the database search. After excluding duplicates, we screened the titles and abstracts of 345 records, and subsequently, 26 articles were retrieved for the full-text assessment. Finally, 14 studies met the inclusion criteria and were included in the systematic review (Figure 3). 

### 3.1. Design and Samples

Of the 14 included studies, seven were described as crossover trials [11,12,13,14,15,16,17] and five as counter-balanced studies [13,17,18,19,20].

In total, 178 male participants were included across the RCTs. The study’s samples ranged from 7 to 22 participants (age range: 16–33 years), with a competitive level varying from amateur to elite. Participants were soccer players in seven studies [11,12,16,21,22,23,24], two studies involved rugby players [8,13], two investigations proposed an intermittent sport-like activity [17,19], and three trials included active people [14,15,20]. 

Regarding the type of exercise period or sports competition carried out just before and after the breaks, two studies performed a soccer match [11,23], five conducted a field-based test [12,17,21,22,24], and seven opted for a laboratory test [13,14,15,16,18,19,20].

### 3.2. Intervention

All RW-U strategies were performed during a break between two exercise phases and lasted between 5 s and 15 min. The majority of the proposed RW-U activities started after 1 to 14 min of passive rest [11,12,13,14,15,16,17,19,20,23,24]. The proposed RW-U routine ranged in intensity from maximum to very low. Passive resting was the most common practice assigned to the control groups [11,12,13,14,15,16,17,18,19,20,21,22,24].

A total of 23 RW-U regimes were examined. Twenty were active and included interventions based on sports movements, speed drills, strength exercises, cycling, running, whole-body vibration, and inspiratory exercises. The passive strategies employed heated jackets or implemented water immersion. All this information is shown in Table 2.

### 3.3. Main Outcomes

We grouped the variables into four broad areas: (1) physiological measures, (2) conditional abilities measures, (3) perceptual measures, and (4) sport efficiency measures.

#### 3.3.1. Physiological Measures

*Heart rate.* The HR measurement was common in 12 studies [11,12,14,15,16,17,19,20,21,22,23,24]. Out of the 11 studies that assessed HR just after the break, ten reported statistically significant effects. In this sense, active interventions were more effective than passive rest [11,12,14,15,17,20,21,22,24] or passive heating [21] for increasing HR. Moreover, the 12 studies assessed the impact of RW-U practices on HR during the second exercise phase. Four of them reported significant increases in this variable and showed that greater intensities of RW-U activities led to higher HR increments [11,14,17,20].*Body temperature.* Nine investigations evaluated the effects of RW-U regimes just after the break ended, and seven of them reported significant effects [13,14,17,18,19,20,21,22,23]. They outlined those active interventions and passive heating-maintained body temperature more than passive rest. Russell et al. [13] showed that the combination of active and passive strategies was more effective than both separately. Two of the six articles that evaluated the impact of RW-U activities on the second exercise phase found significant effects. According to their findings, temperature tended to be higher in cycling protocols than passive rest [14,20].*Gas measurements.* Four investigations measured the impact of RW-U strategies on gas measurements just once the break ended and during the second exercise phase [14,15,20,22]. Two studies found that RW-U practices were better than passive rest for increasing oxygen volume [20,22]. In this regard, [22] observed that gas measurements were higher after repeated sprints than after whole vibration exercises. The performance of RW-U activities appeared to be more effective than passive rest for increasing gas measurements during the second exercise phase in three studies [14,15,20].*Muscle oxygenation*. None of the four studies that assessed the impact of RW-U regimes on muscle oxygenation just after the break ended reported significant effects [14,19,20]. Three investigations evaluated the efficacy of RW-U practice during the second exercise phase [14,19,20], indicating that cycling interventions resulted in greater mean oxygenated hemoglobin values than passive rest.*Blood metabolite response*. Three studies determined the effects of RW-U strategies on blood metabolite response just after the break evaluations [6,19,24], and one study performed additional assessment during the second exercise phase. Zois et al. [16] observed that strength exercises led to significantly lower lactate levels than small-sided games just after the break ended.*Neuromuscular activity (EMG).* One study reported outcomes related to neuromuscular activity just after finish the break. Specifically, Yanaoka et al. [17] showed that running strategies decreased the electromyogram amplitude of maximal voluntary contraction after HT without a maximal voluntary contraction force decrement. Two investigations reported outcomes during the second exercise phase [14,20] where the root mean square was higher in cycling routines when compared with passive rest [14,20]. Moreover, Yanaoka et al. [20] found that the intervention with higher intensity exhibited the highest median frequency results.*Inspiratory muscle function.* Only one study evaluated the impact of RW-U (inspiratory-loaded core exercises) on inspiratory muscle function. Namely, Tong et al. [19] detailed that the inspiratory muscular function was not restored immediately after the break in those athletes who underwent passive rest.

#### 3.3.2. Conditional Abilities Measures

*Sprint performance*. Twelve studies [7,11,12,13,14,15,16,17,18,19,20,23] evaluated sprint performance. Three out of the four studies that assessed the efficacy of RW-U activities on sprint performance just after the break ended detailed that passive rest reduced sprint performance compared with the data obtained in the first exercise phase [2,11,22]. Regarding the impact of RW-U practices on the second exercise phase, statistically significant results were noticed in 10 of the 11 studies. In this aspect, active interventions and passive heating were more effective than passive rest in eight studies [12,13,14,15,17,18,19,20]. Russell et al. [13] found that the combination of active and passive heating strategies was more effective than both separately. On the other hand, Zois et al. [16] showed that strength exercises led to higher sprint performance than small-sided games and passive rest.*Lower body muscular strength*. RW-U activities effectively increased lower body muscular strength just after the break, according to the six studies that assessed this variable [11,12,13,16,18,22]. For example, Russell et al. [13] reported that the combination of active and passive strategies was more effective than both alone. Two studies evaluated lower body muscular strength in the second exercise phase [16,22]. Only Zois et al. [16] detected statistically significant effects; particularly, strength exercises were more effective than small-sided games in improving lower body muscular strength.*Core strength.* One study explored the impact of an RW-U routine (inspiratory-loaded core exercises) on core strength, where Tong et al. [19] informed that participants’ core strength was not restored immediately after the break ended in those who rested passively.*Aerobic endurance*. The two studies that assessed the effects of RW-U strategies in aerobic performance [21,24] found that active interventions (agility sprint drills, running or cycling) led to a significant minor decrease in this variable than passive rest during the second phase.*Anaerobic performance*. No statistically significant effects of an RW-U exercise on anaerobic performance were observed in the only study investigating this variable [19].

#### 3.3.3. Perceptual Measures

*Rating of perceived exertion*. Eight investigations determined the impact of RW-U strategies on the rating of perceived exertion both just after the break ended and during the second exercise phase [12,14,15,16,17,19,20,24]. Six out of the eight studies that carried out evaluations just after the break finished reported significant findings, indicating that active interventions led to a higher rating of perceived exertion than passive rest [14,15,16,17,20,24]. Zois et al. [16] revealed that strength exercises compared with small-sided games led to a higher rating of perceived exertion. Statistically significant effects of RW-U practices on the second exercise phase were identified in two studies, where active interventions resulted in a greater perceived exertion than passive rest [12,15].*Muscle soreness.* Performing RW-U activities did not reduce muscle soreness just after the break ended, according to the results obtained in the only investigation that addressed this topic [12]. Two studies analyzed the impact of RW-U practices on the second exercise phase [12,16]. Only Zois et al. [16] reported significant effects. In comparison with small-sided games and passive rest, strength exercises led to higher muscle soreness.*Rating of perceived breathlessness*. Rating of perceived breathlessness was evaluated in one study [19], in which no significant effects immediately after performing RW-U activities (inspiratory-loaded core exercises) were observed.

#### 3.3.4. Sports Efficiency Measures

On the one hand, only one study examined the impact of conducting an RW-U regime just after the break on variables related to sports efficiency. Zois et al. [16] observed that passing ability was greater in active strategies in comparison with passive rest. On the other hand, two of the three studies that evaluated the effect of RW-U routines on the second exercise phase found significant effects. In this respect, an RW-U exercise was shown to be more effective than passive rest to improve match activities [11] and passing ability [16].

## 4. Discussion

This study aimed to analyze the existing scientific evidence on the effects of RW-U activities on sports performance. Although several studies have been conducted to date, no consensus has been reached on an optimal strategy or protocol to be applied during rest periods in sports matches. This work provides recommendations in this regard, which could be helpful for coaches and players in order to reduce the detrimental effects these rest periods have on sports performance.

Research states that an active warm-up induces more remarkable physiological changes, leading to greater preparation for a subsequent exercise task [25]. Most of the studies reviewed here examined the effects of RW-U practices on HR. In this manner, cycling, running, and calisthenics leads to higher HR values than passive heating or rest, both right after the break and during the second exercise phase. This positive impact of RW-U regimes on HR seems to be related to the activities’ intensity. Despite the significant number of studies that included this variable in their designs, none considered analyzing the impact of using HR variability as biofeedback, an approach that seems to enhance warm-up effects [26].

Muscle temperature is a crucial factor in boosting sports performance. For instance, a large positive association between muscle temperature and power output has been reported, indicating that a 1 °C increase in muscle temperature was accompanied by 2–5% improvement in muscle power performance [27], and every 1 °C decrease was associated with a 3% loss in exercise performance [28]. The results of this systematic review outlined that passive heating and RW-U activities effectively maintained muscle temperature compared to passive rest during the break period. Although combining both strategies seemed to be the best option, only one study addressed this question. Therefore, more research is needed in this regard. According to the reviewed studies, performing cycling during the rest period is also a successful strategy to increase muscle temperature during the second period. These findings further support the idea of Priego-Quesada et al. [29], who claim that after cycling, temperature increases. It would be interesting to explore the effect of this type of intervention, as it applies to many sports, combined with sport-specific exercises [30].

In addition, performing RW-U activities resulted in higher gas measurement values and increased muscle function just after the break was finished. This strategy effectively improved muscular activity and muscle oxygenation during the second period. Altogether, these findings confirm the idea that passive rest diminishes physiological function in athletes, as previously observed, and encourages the inclusion of RW-U routines at half-time.

In line with these findings, the examined studies in this review indicated that passive rest is not the best option for improving conditional skills during the half-match. Performing RW-U activities were positive stimuli for improving sprint performance and muscular strength just after the break. Similarly, both fitness dimensions and aerobic performance improved during the second period after conducting this strategy. Even passive heating exhibited more benefits than resting for enhancing sports performance. The combination of active and passive strategies appeared to be more effective than when performed separately for improving conditional abilities. These results are consistent with other observations suggesting that increases in body temperature are strongly linked to sports performance [31]. 

Moreover, previous research shows that performing lower body strength exercises could effectively improve jumping [32] and sprint performance [33] immediately after interventions. It is encouraging to compare these results with our findings indicating that conditional abilities were greatly enhanced after performing RW-U based on RM exercises compared to small-sided games or passive rest. Nevertheless, it seems challenging to equip sports changing rooms with body-building machines to perform an RW-U routine like those mentioned above.

Data from this review indicated that RW-U led to a higher rate of perceived exertion than passive rest. In addition, they did not reduce muscle soreness just after the break. Similarly, physically demanding activities were perceived as more intense and produced higher muscle soreness scores during the second period. These findings may help to explain why some coaches have reported that avoiding player fatigue is one of the situational factors that may limit practitioners in applying RW-U strategies [34]. However, in the studies included in this review, although the participant’s perceived exertion was high following the RW-U routine, performance improved. Thus, it seems that even though during the half-time break, players tend to be passive [8], it appears that carrying out RW-U activities are beneficial.

The existing literature on the effects of RW-U practices was not only focused on physical variables. Some studies analyzed the impact of these strategies on sport performance measures. In this regard, results were mixed, with two out of three investigations reporting beneficial effects. However, none of the three investigations reported a negative impact, proving a solid justification for team-sport athletes to perform these routines during rest periods. Likewise, the passing ability was enhanced in the small-sided games’ intervention compared to the passive rest and 5RM intervention. The latter suggests that performing sport-specific movements during the RW-U intervention may facilitate transfer to the second period [16]. 

This review had a novel approach as it included RCTs regardless of the level and sport performed to provide a broader view of the subject. Nevertheless, some limitations should be acknowledged. In the first place, the methodological quality of the studies was not high since most of them did not provide enough data to characterize the study sample. Secondly, we found little variety among the sports studied because most of them related to soccer, and no research included female athletes. Thirdly, there was a low ecological validity, as the tests used to assess performance lacked similarity to the competition environment. Finally, the heterogeneity in the intensity and duration of the RW-U activities and the different tasks proposed make it challenging to compare the magnitude of the effect between the RW-U strategies directly.

Based on the results of this review, it seems that at least a short active RW-U is indispensable for improving sports performance, while passive rest should be minimized. Coaches should be aware that performing passive or active strategies are beneficial for keeping warm during rest periods. The combination of both strategies could be more effective. When it comes to designing active RW-U, it is suggested that athletes should execute active strategies based on the performance of sport-specific movements (small-sided games), or strengthening exercises for increasing muscular potentiation, even though it may generate a higher RPE than passive rest. On a final note, it seems that athletes could take advantage of passive strategies (e.g., tracksuit or jacket) while coaches develop the delivery of tactical instructions. In this case, it is advisable to perform muscular potentiation activities during the final minutes of the half-time break.

## 5. Conclusions

A variety of studies have assessed the efficacy of RW-U activities. Scientific evidence indicates that performing these activities during half-time increases physical performance through improvements in both conditional abilities and physiological measures. Thus, coaches should be aware that RW-U activities are a valuable strategy to avoid reducing sports performance during prolonged breaks. According to the findings of this review, athletes should perform short and intense active activities, either oriented towards sport-specific movements, or towards explosive strength exercises, combined with passive RW-U strategies. Future research should identify the impact of RW-U practices in different team sports and include actual or simulated competitions in their analysis.

## Figures and Tables

**Figure 1 medicina-57-00976-f001:**
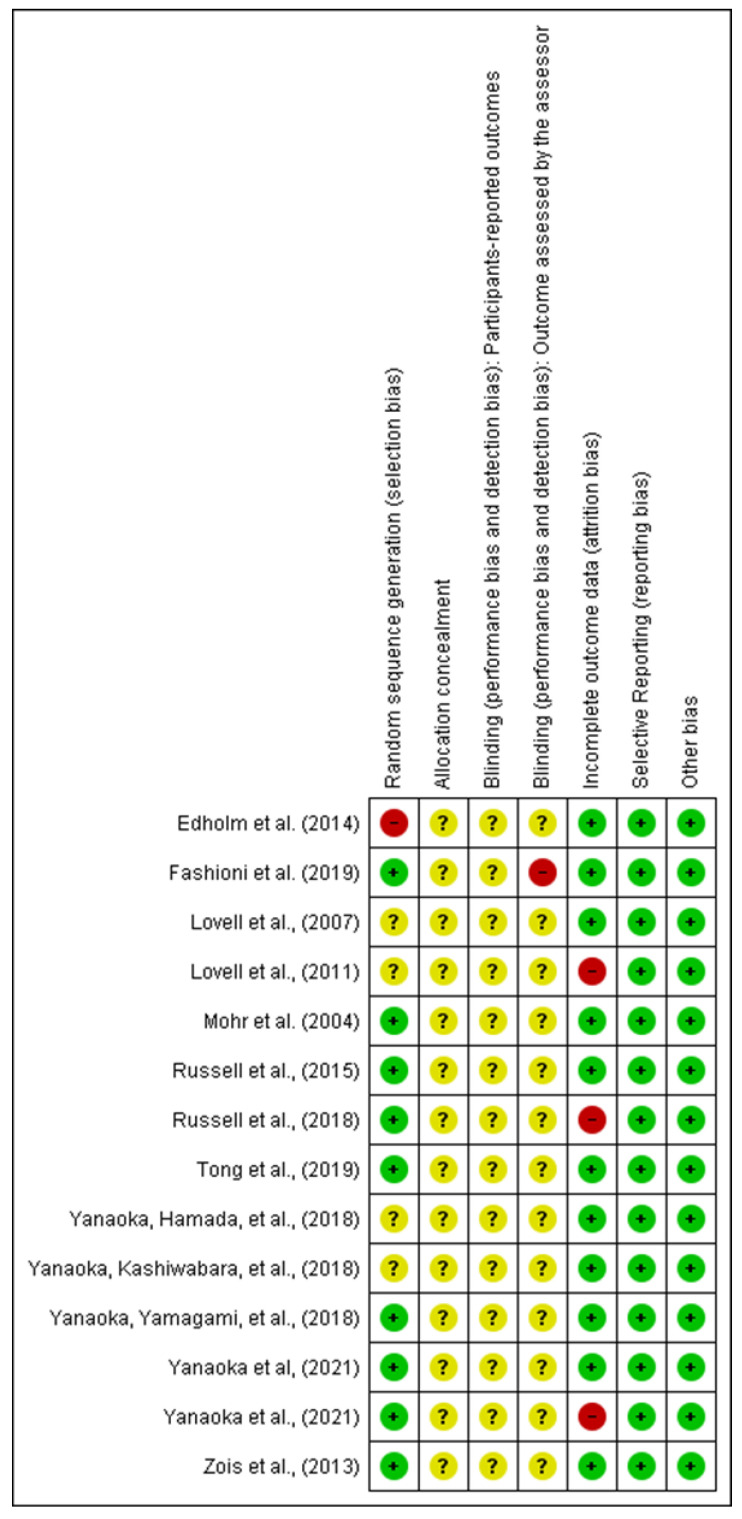
Review authors’ judgements about each risk of bias item for each included study.

**Figure 2 medicina-57-00976-f002:**
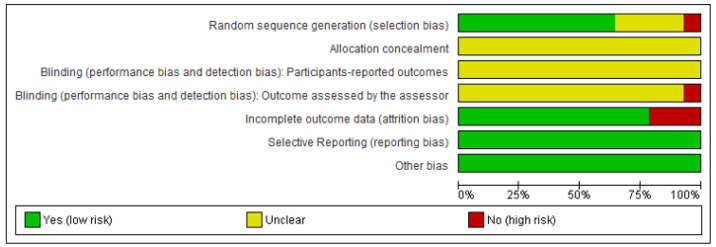
Review authors’ judgements about each risk of bias item presented as percentages.

**Figure 3 medicina-57-00976-f003:**
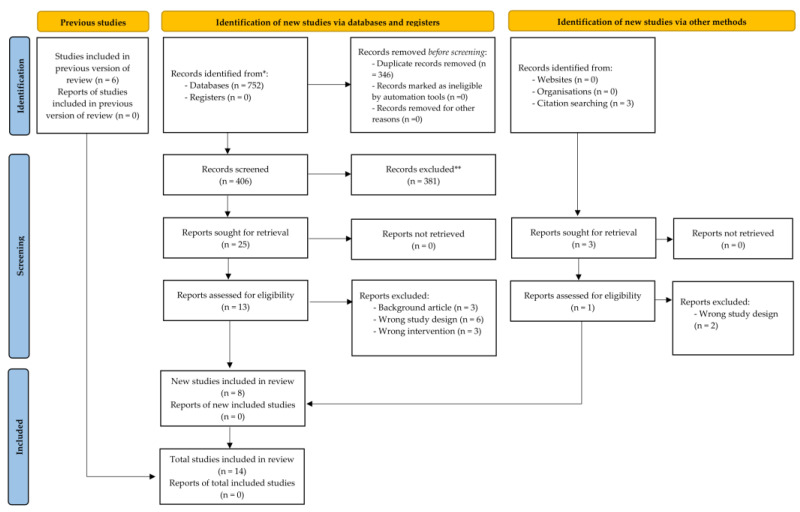
PRISMA (Preferred Reporting Items for Systematic Reviews and Meta-Analyses) study flow diagram.

**Table 1 medicina-57-00976-t001:** Search strategy and inclusion/exclusion criteria based on PICO (Population, Intervention, Comparison and Outcome).

Databases	Search Terms	PICO	Inclusion Criteria	Exclusion Criteria
Web of Science Scopus PubMed SPORTDiscus	“Re-warm-up” OR “Half-time strategy” OR “Second-half” AND “warm-up” OR “RW-U strategy”	Population	Sport athletes and healthy people	-
		Intervention	Re-warm-up (active or passive) strategies during half-time	Not considered by the researchers a practical method to apply to sports The interventions were not carried out in neutral environments
		Comparison	Re-warm-up strategies/conditions	No comparison between structured strategies or control condition with pre/post-results
		Outcome	Physiological measures Conditional abilities measures Perceptual measures Sport efficiency measures	The outcomes did not consider physical or technical-tactical measures They lacked data regarding the effects of RW-U routines during an exercise period or sports competition carried out immediately after a break

**Table 2 medicina-57-00976-t002:** Descriptive characteristics of the studies that proposed re-warm-up interventions during half-time.

First Author (Year), Design and Participants	Intervention	Outcomes (Test)	Significant Effects	
			Acute Changes	Short-Term Changes
**Edholm et al. (2014)** **Design**: RCT **Participants**: Soccer players *Level*: Professional (top league in Sweden, Allsvenskan) *Category*: Senior *Sample* (n; sex): 22 M Age, years (mean; range): 25; 18–33 Stature, cm (mean; range): 182; 175–195 Body mass, kg (mean; range): 78.6; 69.3–93.6	**Soccer match simulation (90 min)** *P1* (1st half, 45 min) *P2* (2nd half, 45 min) **Half-time (15 min)** *G1* (7 min passive rest + 7 min × jogging and light calisthenics at low-moderate intensity, i.e., 70% of HR_max_) *CON* (15 min × passive seated rest, with rehydration and coaching instructions)	**Physiological measures** *Heart rate* *Weight loss* **Conditional abilities** *Lower body muscular strength* (CMJ) *Sprint performance* (10 m sprint) **Sport efficiency measures** *Match activities* (Distance covered; Technical skill; Defensive and offensive high intensity runs; MEPT; Ball possession)	**Intra-group (*p* < 0.05, *pre-P2*** vs. ***P1*****)** CMJ ↓ in *G1* (37.5 ± 3.7 vs. 38.7 ± 3.7 cm) CMJ ↓ in *CON* (36.4 ± 3.9 vs. 39.0 ± 2.9 cm) Sprint performance ↓ in *CON* (1.98 ± 0.06 vs. 1.93 ± 0.05 s) **Inter-group (*p* < 0.05, *pre-P2*)** >mean HR in *G1* (117 ± 10 bpm) than in *CON* (109 ± 12 bpm) >CMJ ↓ in *CON* (7.6%; 36.4 ± 3.9 cm) than in *G1* (3.1%; 37.5 ± 3.7 cm)	**Intra-group (*p* < 0.05, *P2*** vs. ***P1*****)** ↓ mean HR in *G1* (157 ± 12 vs. 167 ± 7 bpm) and *CON* (161 ± 11 vs. 167 ± 8 bpm) ↓ total distance covered in *G1* (0.16 ± 0.01 vs. 0.17 ± 0.02 m per MEPT) and *CON* (0.17 ± 0.01 vs. 0.19 ± 0.02 m per MEPT) ↓ defensive high-intensity distance during first 15 min of each half in *G1* (0.14 ± 0.06 vs. 0.21 ± 0.07 km) ↑ ball possession during first 5 and 15 min of each half in *G1* ↓ number of passes in *CON* (100 ± 4 vs. 113 ± 13) ↓ number of dribbles in CON (20 ± 2 vs. 26 ± 11) **Inter-group (*p* < 0.05, *P2*)** <distance covered during first third in *G1 (9%)* than in *CON (4%)* <number of occurrences of high intensity runs (3.3 ± 0.7 vs. 3.8 ± 1.3 times per MEPT) in *G1* than in *CON* <number of occurrences of sprinting (0.5 ± 0.2 vs. 0.6 ± 0.2 times per MEPT) in *G1* than in *CON*
**Fashioni et al. (2020)** **Design**: RCT **Participants**: Soccer players *Level*: Amateur *Category*: NR *Sample* (n; sex): 10 M Age, years (mean ± SD): 23 ± 4 Stature, cm (mean ± SD): 182.0 ± 6.4 Body mass, kg (mean ± SD): 77.3 ± 7.2	**Field-based test-SAFT90 (90 min)** *P1* (SAFT90, 45 min) *P2* (SAFT90, 45 min) **Half-time (15 min)** *G1* (12 min × passive rest + 3 min × combination of bodyweight exercises and ballistic and plyometric movements, intensity NR) *CON* (15 min × passive seated rest, water ad libitum)	**Physiological measures** *Heart Rate* **Conditional abilities** *Lower body muscular strength* (CMJ; SJ) *Sprint performance* (Speed of 5, 10, and 20 m) **Perceptual measures** *Muscle soreness* (VAS) *Rating of perceived exertion–RPE* (Borg scale 6–20) **Sport efficiency measures** *Player-load metrics* (PLtotal; PLML; PLAP; PLV)	**Inter-group (*p* < 0.05, *pre-P2*)** >mean HR in *G1* (160 ± 14 bpm) than in *CON* (93 ± 9 bpm) >CMJ in *G1* (31.5 ± 5.4 cm) than in *CON* (28.2 ± 4.7 cm) >SJ in *G1* (30.2 ± 5.1 cm) than in *CON* (27.0 ± 5.0 cm)	**Inter-group (*p* < 0.05, *P2*)** <20 m sprint in *G1* (3.32 ± 0.12 s) than in CON (3.42 ± 0.20 s) >RPE at 50 min in *G1* (13 ± 2 a.u) than in *CON* (11 ± 1 a.u) >RPE at 55 min in *G1* (14 ± 2 a.u) than in *CON* (12 ± 1 a.u) >RPE at 60 min in *G1* (14 ± 2 a.u) than in *CON* (13 ± 1 a.u)
**Lovell et al. (2007)** **Design**: RCT **Participants**: Soccer players *Level*: Elite (English League One) *Category*: Senior *Sample* (n; sex): 7 M Age, years (mean ± SD): 17.4 ± 0.5 Stature: NR Body mass, kg (mean ± SD): 69.1 ± 3.1	**Field-based test-Bangsbo field test (90 min)** *P1* (Bangsbo field test, 45 min) *P2* (Bangsbo field test, 45 min) **Half-time (15 min)** *G1* (7–14 min × passive heating, immersed in 40°C water up to the gluteal fold) *G2* (7–14 min × cycle ergometer at 70% HR_max_) *G3* (7–14 min × repeated agility sprint drills, i.e., sprinting, bounding, jumping and other utility movements common to soccer; at 70% HR_max_) *CON* (15 min × passive rest)	**Physiological measures** *Heart Rate* *Body temperature* (Tc) *Weight loss* **Conditional abilities** *Aerobic endurance* (Bangsbo field test)	**Intra-group (*p* ≤ 0.05, *pre-P2*** vs. ***P1*****)**: Tc ↓ in *G2* (0.52 ± 0.18 °C) and *CON* (0.97 ± 0.29 °C) **Inter-group changes (*p* ≤ 0.05, *pre-P2*)**: >mean HR in *G2* (128 ± 5 bpm) and *G3* (128 ± 8 bpm) than in *CON* (110 ± 4 bpm) and *G1* (113 ± 6 bpm) >mean HR ↓ in *G3* than in *G2* <Tc ↓ in *G2* (0.52 ± 0.18 °C) than in *G1* (0.7 ± 0.4 °C), *G3* (0.77 ± 0.16 °C) and *CON* (0.97± 0.29 °C)	**Intra-group changes (*p* ≤ 0.05, *P2*** vs. ***P1*****)**: Bangsbo field test (distance) ↓ in *CON* (3.1 ± 1.9%) **Inter-group changes (*p* ≤ 0.05, *P2)***: <Bangsbo field test (distance) ↓ in *G2* (−0.5 ± 1.3%) and *G3* (−0.4 ± 1.4%) than in *CON*
**Lovell et al. (2011)** **Design**: RCT **Participants**: Soccer players *Level*: Semi-professional *Category*: NR *Sample* (n; sex): 10 M Age, years (mean ± SD): 20 ± 1 Stature, cm (mean ± SD): 183.0 ± 9 Body mass, kg (mean ± SD): 79.9 ± 7.0	**Field-based test-SAFT90 (90 min)** *P1* (SAFT90, 45 min) *P2* (SAFT90, 45 min) **Half-time (15 min)** *G1* (9–14 min × repeated 20-m soccer-specific runs at moderate to high speed, i.e., 70% HR_max_) *G2* (9–14 min × intermittent exposure to WBV, intensity NR) *CON* (15 min × passive rest)	**Physiological measures** *Heart Rate* *Gas measurements* (VO_2_) *Body temperature* (Tm) **Conditional abilities** *Lower body muscular strength* (CMJ; CQ; CH; EH) *Sprint performance* (10 m sprint)	**Intra-group (*p* < 0.05, *pre-P2*** vs. ***P1*****)** CMJ, EH and sprint performance ↓ in *CON* **Inter-group (*p* < 0.05, *pre-P2*)** >mean HR in *G1* (140± 6 bpm) than in *G2* (104 ± 11 bpm) and *CON* (92 ± 13 bpm) >VO_2_ (mL/kg/min) in *G1* (31.8 ± 4.8) and *G2* (10.8 ± 2.8) than in *CON* (6.3 ± 1.9) >CMJ in *G1* and *G2* than in *CON* <Tm ↓ in *G1* than in *G2* and *CON* <CH peak torque in *G1* than in *CON*	
**Mohr et al. (2004)** **Design**: RCT **Participants**: Soccer players *Level*: Semiprofessional (Danish Fourth Division) *Category*: Senior *Sample* (n; sex): G1: 9 M; G2a: 8 M; G2b: 8 M Age, years (mean ± SEM): G1: 27.0 ± 1.5; G2a: 26.0 ± 0.5; G2b: 25.8 ± 1.4 Stature, cm (mean ± SEM): G1: 181.0 ± 1.8; G2a: 181.8 ± 1.8; G2b: 183.1 ± 1.9 Body mass, kg (mean ± SEM): G1: 81.1 ± 1.2; G2a: 76.8 ± 3.3; G2b: 76.2 ± 1.5	**Soccer friendly match (90 min)** *P1* (1st half, 45 min) *P2* (2nd half, 45 min) **Half-time (15 min)** *G1* (7 min × passive rest + 7 min × running and other exercises at moderate intensity, i.e., HR 135 bpm or 70% of the peak HR reached during the match) *CON* (10 min × passive rest + 5 min physical activities at very low intensity)	**Physiological measures** *Heart Rate* *Body temperature* (Tm; Tc) *Weight loss* **Conditional abilities** *Sprint performance* (3 × 30 m sprint)	**Intra-group (*p* < 0.05, *pre-P2*** vs. ***P1*****)** ↓ in Tm (37.7 ± 0.2 vs. 39.1 ± 0.2 and 39.7 ± 0.2 °C), Tc (37.8 ± 0.1 vs. 38.2 ± 0.1 °C) and sprint performance (2.4 ± 0.3%) in *CON* **Inter-group (*p* < 0.05, *pre-P2*)** >Tm in *G1* (39.2 ± 0.2 °C) than in *CON* (37.7 ± 0.2 °C)	**Intra-group (*p* < 0.05, *P2*** vs. ***P1*****)** ↓ mean sprint performance (2.3 ± 0.3%) in *G1*
**Russell et al. (2015)** **Design**: RCT **Participants**: Rugby players *Level*: Professional (French top tier) *Category*: Senior *Sample* (n; sex): 18 M Age, years (mean ± SD): 23 ± 1 Stature, cm (mean ± SD): 183 ± 5 Body mass, kg (mean ± SD): 96.4 ± 8.7	**Laboratory test-****RSSA** *P1* (RSSA) *P2* (RSSA) **Half-time (15 min)** *G1* (15 min × passive rest with a survival jacket) *CON* (15 min × passive rest)	**Physiological measures** *Body Temperature* (Tc) **Conditional abilities** *Lower body muscular strength* (CMJ, i.e., peak power output) *Sprint performance* (RSSA, i.e., best sprint, mean sprint and total sprint)	**Inter-group (*p* ≤ 0.05, *pre-P2*)** <Tc ↓ in *G1* (0.74 ± 0.08%) than in *CON* (−1.54 ± 0.06%) >peak power output in *G1* (5610 ± 105 W) than in *CON* (5440 ± 105 W)	**Inter-group (*p* ≤ 0.05, *P2*)** <best sprint time in *G1* than in *CON* (1.39 ± 0.17%) <mean sprint time in *G1* than in *CON* (0.55 ± 0.06%) <total sprint time in *G1* than in *CON* (0.55 ± 0.06%)
**Russell et al. (2018)** **Design**: RCT **Participants**: Rugby players *Level*: Professional (French top tier) *Category*: Senior *Sample* (n; sex): 20 M Age, years (mean ± SD): 24 ± 5 Stature, cm (mean ± SD): 185 ± 1 Body mass, kg (mean ± SD): 97.5 ± 7.8	**Laboratory test-****RSSA** *P1* (RSSA) *P2* (RSSA) **Half-time (15 min)** *G1* (15 min × passive rest with a survival jacket) *G2* (8 min passive rest + 7 min × jogging and simple ball skills at low-medium intensity, i.e., mean HR of 136 ± 4 bpm) *G3* (8 min × wearing a survival jacket + 7 min × jogging and simple ball skills at low-medium intensity, i.e., mean HR of 136 ± 4 bpm) *CON* (15 min × passive rest)	**Physiological measures** *Body Temperature* (Tc) **Conditional abilities** *Lower body muscular strength* (CMJ, i.e. peak power output) *Sprint performance* (RSSA)	**Inter-group (*p* ≤ 0.05, *pre-P2*)** <Tc ↓ in *G1* (−0.23 ± 0.09 °C), *G2* (−0.17 ± 0.09 °C) and *G3* (−0.03 ± 0.10 °C) than in *CON* (*0.62± 0.28* °C) <peak power output ↓ in *G1* (−213 ± 79 W), G2 (−83 ± 72 W) and G3 (10 ± 52 W) than in *CON* (−385 ± 137 W)	**Inter-group (*p* ≤ 0.05, *P2*)** ↑ Sprint performance in *G3* (6.74 ± 0.21 s) *G1* (6.82± 0.04 s) and *G2* (6.80± 0.05 s) than in *CON* (6.85± 0.04 s) >sprint performance in *G1* (6.82 ± 0.04 s) and *G2* (6.80 ±0.05 s) than in *CON* (6.85 ± 0.04 s)
**Tong et al. (2019)** **Design**: RCT **Participants**: Soccer and handball players *Level*: College *Category*: NR *Sample* (n; sex): 9 M Age, years (mean ± SD): 20.6 ± 0.9 Stature, cm (mean ± SD): 174 ± 6 Body mass, kg (mean ± SD): 68.8 ± 8.8	**Laboratory test-IEP** *P1* (IEP, 25.8 min) *P2* (IEP, 7.5 min) **Half-time (15 min)** *G1* (11 min passive rest + 4 min × 4 inspiratory-loaded CM exercises) *CON* (15 min × passive rest)	**Physiological measures** *Heart Rate* *Muscle oxygenation* (Oxy-Hb; Deoxy-Hb; Total-Hb) *Skin temperature* (Ts) *Blood Metabolite Response* ([La]) *Inspiratory muscular function* (PImax) **Conditional abilities** *Sprint performance* (RSA, i.e., peak velocity, mean velocity, acceleration) *Anaerobic performance* (IEP) *Core muscular strength* (SEPT) **Perceptual measures** *Rating of perceived exertion* (Borg scale 6–20) *Ratings of perceived breathlessness* (Borg scale 0–10)	**Intra-group (*p* ≤ 0.05, *pre-P2*** vs. ***P1*****)** PImax ↓ in *CON* (−6.4%) SEPT ↓ in *CON* (−19.0%) **Inter-group (*p* ≤ 0.05, *pre-P2*)** Ts returned to the Pre-P1 level in *CON* (31.9 ± 0.5 °C), but not in *G1* (30.4 ± 0.5 °C)	**Intra-group (*p*****≤****0.05, *P2*** vs. ***P1*****)** Peak velocity ↑ (3.0%) in *G1* Mean velocity ↑ (2.0%) in *G1* **Inter-group (*p* ≤ 0.05, *P2*)** >peak velocity in *G1* (0.15 ± 0.006) than in *CON* (0.13 ± 0.08) >mean velocity in *G1* (0.09 ± 0.001) than in *CON* (−0.1 ± 0.09)
**Yanaoka, Yamagami et al. (2018)** **Design**: RCT **Participants**: Soccer referees *Level*: 2nd, 3rd or 4th class registered official licenses (Japan Football Association) *Category*: NR *Sample* (n; sex): 10 M Age, ***years*** (mean ± SD): 22 ± 1 Stature, ***cm*** (mean ± SD): 173.6 ± 5.8 Body mass, ***kg*** (mean ± SD): 67.2 ± 6.4	**Field-based tests–LIST and Yo-Yo IR1** *P1* (LIST, 45 min) *P2* (Yo-Yo IR1) **Half-time (15 min)** *G1* (13 min × seated rest on a chair for 2 min 15 s + running for 2 min 15 s at 70% HR_max_, that were successively repeated; beginning 1 min after the start of the HT period and finished 1 min prior to beginning the Yo-Yo IR1) *CON* (15 min × passive rest)	**Physiological measures** *Heart Rate* *Blood Metabolite Response* (Plasma glucose; FFA; TG; CK; [La]) **Conditional abilities** *Aerobic Endurance* (Yo-Yo IR1) **Perceptual measures** *Rating of perceived exertion-RPE* (Borg scale 6–20)	**Intra-group (*p* < 0.05, *pre-P2*** vs. ***P1*****)** RPE ↓ in *CON* **Inter-group (*p* < 0.05, *pre-P2*)** >mean HR in *G1* (105 ± 10 bpm) than in *CON* (82 ± 8 bpm) >RPE in *G1* than in *CON*	**Inter-group (*p* < 0.05, *P2*)** >Yo-Yo IR1 performance in *G1* (3.095 ± 326 m) than in *CON* (2.904 ± 421 m)
**Yanaoka, Hamada et al. (2018)** **Design**: RCT **Participants**: healthy men who trained (i.e.,, more than an hour/session) for more than 2 days/week *Level*: NA *Category*: NA *Sample* (n; sex): 11 M Age, years (mean ± SD): 22.7 ± 2.4 Stature, cm (mean ± SD): 173 ± 6 Body mass, kg (mean ± SD): 65.3 ± 10.0	*P1* (Cycling intermittent exercises, 40 min) *P2* (Laboratory test- CISP, 20 min) **Half-time (15 min)** *G1* (11 min × passive rest + 3 min × cycle ergometer at 60% of VO_2max_; ending 1 min before the start of the CISP) G2 (11 min × passive rest + 3 min × cycle ergometer at 30% of VO_2max_; ending 1 min before the start of the CISP) *CON* (15 min × rest on the cycle ergometer)	**Physiological measures** *Heart Rate* *Gas measurements* (VO_2_, VCO_2_, RER) *Muscle oxygenation* (Oxy-Hb, Deoxy-Hb, Total-Hb, SmO_2_) *Body Temperature* (Ts, Tm) *Neuromuscular Activity-EMG* (RMS) **Conditional abilities** *Sprint performance* (CISP) **Perceptual measures** *Rating of perceived exertion–RPE* (Borg scale 6–20)	**Inter-group (*p* ≤ 0.05, *pre-P2*)** >mean HR (%HR_max_) in *G1* (63 ± 7) ) and *G2* (52 ± 3) than in *CON* (46 ± 5) >RPE in *G1* (10.8 ± 1.5 a.u) and *G2* (11.8 ±1.7 a.u) than in *CON* (8.2 ± 1.7 a.u)	**Inter-group (*p* ≤ 0.05, *P2*)** >mean HR (%HR_max_) in *G1* (77 ± 5) than in *CON* (72 ± 4) >mean VCO_2_ (0.1–5.0 mL/kg/min) in *G1* than in *CON* >mean RER (0.01–0.08) in *G1* than in *CON* >Δoxy-Hb in *G1* (1.8 ± 2.0 umol/L) and *G2* (1.0 ± 4.3 umol/L) than in *CON* (−2.0 ± 3.9 umol/L) >Tm in *G1* than in *CON* (CI: 0.4–2 °C) and *G2* (CI: 0.2–1.5 °C) at 10 min of the *P2* >Tm in *G1* than in *CON* (CI: 0.1–1.5 °C) and *G2* (CI: 0.1–1.1 °C) at 15 min of the *P2* >Mean RMS in *G1* (CI: 0.2–23.2%) and *G2* (CI: 0.5–35.0%) than in *CON* >Sprint performance G*1* (CI: 73–490 J) and *G2* (CI: 8–325 J) than *CON*
**Yanaoka, Kashiwabara et al. (2018)** **Design**: RCT **Participants**: healthy men who trained (i.e., more than an hour/session) for more than 2 days/week *Level*: NA *Category*: NA *Sample* (n; sex): 13 M Age, years (mean ± SD): 22.4 ± 2.1 Stature, cm (mean ± SD): 172 ± 5 Body mass, kg (mean ± SD): 67.0 ± 10.1	*P1* (Cycling intermittent exercises, 40 min) *P2* (Laboratory test- CISP, 20 min) **Half-time (15 min)** *G1* (8 min × passive rest + 7 min × cycle ergometer at 70% of HR_max_) *G2* (12 min × passive rest + 3 min × cycle ergometer at 70% of HR_max_) *CON* (15 min × passive rest)	**Physiological measures** *Heart Rate* *Gas measurements* (VO_2_, VCO_2_, RER) *Muscle oxygenation* (Oxy-Hb, Deoxy-Hb, Total-Hb, SmO_2_) **Conditional abilities** *Sprint performance* (CISP) **Perceptual measures** *Rating of perceived exertion–RPE* (Borg scale 6–20)	**Inter-group (*p* ≤ 0.05, *pre-P2*)** >mean HR (%HR_max_) in *G1* (66 ± 8) and *G2* (63 ± 6) than in *CON* (48 ± 5) >RPE in *G1* (11.8 ± 1.7 a.u) and *G2* (10.8 ± 1.5 a.u) than in *CON* (8.2 ± 1.7 a.u)	**Inter-group (*p* ≤ 0.05, *P2*)** >VO_2_ (ml/kg/min) in *G1* (29.2 ± 0.8) and *G2* (29.5 ± 0.9) than in *CON* (27.1 ± 1.2) >VCO_2_ (ml/kg/min) in *G1* (27.6 ± 0.8) and *G2* (27.9 ± 0.9) than in *CON* (24.7 ± 1.3) >RER in *G1* (0.95 ± 0.02) and *G2* (0.95 ± 0.02) than in *CON* (0.91 ± 0.02) >Δoxy-Hb in *G1* (−0.6 ± 6.8 umol/L) and *G2* (0.1 ± 3.5 umol/L) than in *CON* (−2.5 ± 3.7 umol/L) >sprint performance in *G1* (3808 ± 949 J) and *G2* (3827 ± 960 J) than in *CON* (3638 ± 906 J) >RPE in *G1* (13.2 ± 1.2 a.u) than in *CON* (12.2 ± 1.5 a.u)
**Yanaoka et al. (2020)** **Design**: RCT **Participants**: active males who habitually exercised for more than 2 days/week *Level*: NA *Category*: NA *Sample* (n; sex): 12 M Age, years (mean ± SD): 23 ± 2 Stature, cm (mean ± SD): 171 ± 5 Body mass, kg (mean ± SD): 68.5 ± 8.7	*P1* (Cycling intermittent exercises, 40 min) *P2* (Laboratory test- CISP, 10 min) **Half-time (15 min)** *G1* (11 min × passive rest + 3 min × cycle ergometer at 30% of VO_2max_; ending 1 min before the start of the CISP) *G2* (13 min × passive rest + 1 min × cycle ergometer at 90% of VO_2max_; ending 1 min before the start of the CISP) *CON* (15 min × rest on the cycle ergometer)	**Physiological measures** *Heart Rate* *Gas measurements* (VO_2_, VCO_2_, RER) *Muscle oxygenation* (Oxy-Hb, Deoxy-Hb, Total-Hb, SmO_2_) *Body Temperature* (Tr, Ts, Tm) *Neuromuscular Activity-EMG* (RMS, MDF, MVC) **Conditional abilities** *Sprint performance* (CISP) **Perceptual measures** *Rating of perceived exertion–RPE* (Borg scale 6–20)	**Inter-group (*p* ≤ 0.05, *pre-P2*)** >mean HR (%HR_max_) in *G1* (49 ± 5) and *G2* (68 ± 4) than in *CON* (46 ± 5) >mean VO_2_ in *G1* and *G2* than in *CON* >RPE in *G2* (11.8 ± 2.1 a.u) than in *G1* (10.4 ± 2.0 a.u) and *CON* (9.5 ± 2.4 a.u)	**Inter-group (*p* ≤ 0.05, *P2*)** >mean HR (%HR_max_) in *G2* (74 ± 6) than in *G1* (71 ± 4) and *CON* (70 ± 5) >mean VO_2_ in *G1* and *G2* than in *CON* >mean VCO_2_ in *G1* than in *CON* >mean RER in *G1* than in *CON* >mean Δoxy-Hb in *G2* than in *CON* >mean Δdeoxy-Hb in *G1* than in *CON* >mean Δtotal-Hb in *G1* and G2 than in *CON* >mean Ts in *G1* (34.2 ± 1.0 °C) and G2 (34.2 ± 1.2 °C) than in *CON* (33.3 ± 1.2 °C) >mean Tm in *G1* (36.3 ± 1.1 °C) and G2 (36.5 ± 1.0 °C) than in *CON* (35.5 ± 1.0 °C) >RMS in *G1* than in *CON* >MDF in *G2* than in *G1* and *CON* >sprint performance in *G1* (3724 ± 720 J) and *G2* (3739 ± 736 J) than in *CON* (3539 ± 698 J)
**Yanaoka et al. (2021)** **Design**: RCT **Participants**: university-based population, ≥5 years of intermittent team sports experience (soccer, basketball, handball, and lacrosse) *Level*: Amateur *Category*: NA *Sample* (n; sex): 12 M Age, years (mean ± SD): 22 ± 2 Stature, cm (mean ± SD): 170 ± 8 Body mass, kg (mean ± SD): 65.1 ± 8.3	**Field-based test–LIST** *P1* (LIST, 45 min) *P2* (LIST, 45 min) **Half-time (15 min)** *G1* (14 min × passive rest + 1 min × running at high intensity, i.e., 90% VO_2max_) *CON* (15 min × passive rest)	**Physiological measures** *Heart Rate* *Body Temperature* (Tga) *Neuromuscular Activity-EMG* (MVC, iEMG, NME) **Conditional abilities** *Sprint performance* (LIST) **Perceptual measures** *Rating of perceived exertion–RPE* (11-point scale)	**Inter-group (*p* ≤ 0.05, *pre-P2*)** >HR in *G1* than in *CON* >Tga in *G1* (38.0 ± 0.4 °C) than in *CON* (37.7 ± 0.3 °C) >iEMG in *G1* (83 ± 5%) than in *CON* (88 ± 12%) >NME in *G1 (*110 ± 14%) than in *CON (*107 ± 14%) >RPE in *CON (*5.2 ± 1.3 au) than in *G1 (*6.1 ± 1.2 au)	**Intra-group changes (*p* < 0.05, *P2*** vs. ***P1*****)**: Mean sprint performance ↓ in *CON (CI: 0.3–6.1%)* Inter-group changes (*p* < 0.05, *P2*): >HR in *G1* than in *CON* >mean sprint performance in *G1* (CI: 1.3–3.4%) than in *CON*
**Zois et al. (2013)** **Design**: RCT **Participants**: Soccer players *Level*: Amateur (division one of the Victorian Football Federation, Australia) *Category*: Senior *Sample* (n; sex): 8 M Age, years (mean ± SD): 23.6 ± 4.1 Stature, cm (mean ± SD): 173.0 ± 5.2 cm Body mass, kg (mean ± SD): 75.5 ± 7.0	**Laboratory test–IAP** *P1* (IAP, 26 min) *P2* (IAP, 26 min) **Half-time (15 min)** *G1* (10 min × passive rest + ~15 s × 5RM performed on a 45° seated leg-press at maximal intensity; ending 4 min before the start of the IAP) *G2* (8 min × passive rest + 3 min × SSG of 2 vs. 2, ball-possession game on a 20-m × 12-m field, intensity NR; ending 4 min before the start of the IAP) *CON* (15 min × passive rest)	**Physiological measures** *Heart Rate* *Blood Metabolite Response* ([La]) **Conditional abilities** *Lower body muscular strength* (CMJ, i.e., flight-time to contraction time ratio, peak velocity, relative-maximum rate of-force development; 5RM leg-press test) *Sprint performance* (RSA, i.e., peak velocity, mean velocity, acceleration) **Perceptual measures** *Muscle Soreness*–*MS* (VAS) *Rating of perceived exertion*–*RPE* (CR0–10) Sport efficiency measures *Skill performance-*LSPT	**Intra-group (*p* ≤ 0.05, *pre-P2*** vs. ***P1*****)** LSPT performance ↑ (6.4%) in *G2* LSPT performance ↓ (7.3%) in *CON* Inter-group (*p* ≤ 0.05, *pre-P2*) <HR in *G1* than in *CON* (28.4%) <[La] in *G1* (3.6 mmol/L) than in *G2* (7.2 mmol/L) >CMJ flight-time to contraction-time ratio in *G1* than in *G2* (9.8%, ES: 0.5 ± 0.3) and *CON* (9.4%, ES: 0.7 ± 0.5) >RPE in *G1* than in *G2* (31.3%, ES: 0.8 ± 0.4) >LSPT performance in *G1* (17.7%, ES: 1.2 ± 0.8) and *G2* (14.7%, ES: 1.7 ± 0.8) than in *CON*	**Intra-group (*p* ≤ 0.05, *P2* vs. *P1*)** Peak velocity ↑ (4.6%) in *G1* Mean velocity ↑ (3%) in *G1* Acceleration in ↑ (18%) *G1* LSPT performance ↑ (6.2%) in *G2* LSPT performance ↓ (9.9%) in *CON* Inter-group (*p* ≤ 0.05, *P2*) >CMJ flight-time to contraction-time ratio in *G1* than in *G2* (8.8%, ES: 0.5 ± 0.3) and *CON* (10.2%, ES: 0.6 ± 0.6) >CMJ peak velocity in *G1* (3%, ES: 0.4 ± 0.3) and *G2* (2.4%, ES: 0.3 ± 0.2) than in *CON* >CMJ relative-maximum rate of-force development in *G1* than in *G2* (29.3%, ES: 0.7 ± 0.5) and *CON* (16.2%, ES: 0.6 ± 0.6) >MS in *G1* than in *G2* (39.5%, ES: 0.7 ± 0.7) and *CON* (49.7%, ES: 0.7 ± 0.7) >RPE in *G1* (29%, ES: 0.8 ± 0.5) and *G2* (22%, ES: 0.5 ± 0.5) than in *CON* >LSPT performance in *G1* (17.2%, ES: 1.5 ± 0.6) and *G2* (12.4%, ES: 0.7 ± 0.7) than in *CON*

>: Greater; <: Lower; ↑: Increment; ↓: Decrement; [La]: Blood lactate; CH: Concentric hamstring; CISP: Cycling Intermittent-Sprint Protocol; CK: creatine kinase; CM: Core muscle; CMJ: Counter movement jump; CQ: Concentric quadriceps; Deoxy-Hb: Deoxygenated hemoglobin; EH: Eccentric hamstring; EMG: Electromyograms; G1: Experimental group; FFA: Free fatty acids; HR: Heart Rate; HT: Half-time; IAP: Intermittent activity protocol; iEMG: Integrated electromyogram; IEP: Intermittent exercise protocol; LIST: Loughborough Intermittent Shuttle Test; LSPT: Loughborough Soccer Passing Test; M: Male; MDF: Median frequency; MEPT: Match activities adjusted for effective playing time; MS: Muscle soreness; MVC: Maximum voluntary isometric contraction; NME: neuromuscular efficiency; NR: Not reported; Oxy-Hb: Oxygenated hemoglobin; P1: Period 1; P2: Period 2; PImax: Maximal inspiratory pressure; PLAP: Anteroposterior movement planes; PLML: Medial-lateral movement planes; PLtotal: Triaxial movement planes; PLV: Vertical movement planes; RCT: Randomized controlled trial; RER: Respiratory exchange ratio; RMS: Root mean square; RPE: Rating of perceived exertion; RSA: Repeated-sprint ability; RSSA: Repeated sprint test; SAFT90: Soccer Aerobic Field Test; SEPT: Sport-specific endurance plank test; SJ: Squat jump; SmO_2_: Muscle oxygen saturation; SSG: Small-sided game; Tc: Core temperature; Tga: gastrointestinal temperature; TG: Serum triglycerides; Tm: Muscle temperature; Total-Hb: Total hemoglobin; Tr: Rectal temperature; Ts: Skin temperature; VAS: Visual analogue scale; VCO_2_: Carbon dioxide volume; VO_2_: Oxygen volume; WBV: Whole-body vibration; Yo-Yo IR1: Yo-Yo Intermittent Recovery Test level 1.

## Data Availability

Not applicable.

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
