# Peer review of "The Efficacy of Re-Warm-Up Practices during Half-Time: A Systematic Review"

_medicina, 2021, doi:10.3390/medicina57090976_

Round 1

Reviewer 1 Report

General Comments

Thank you for the opportunity to review your paper. This paper was conducted to “review and critically analyze the results of those Randomized Control Trials (RCTs) that provided information on the effects of re-warm-up on parameters related to sports performance, regardless of the sport modality investigated and the athletes' level”. In general, the topic of the paper is interesting and fits the scope of the journal. However, there are several methodological issues that need to be clarified prior to its acceptance, of which the most important one is that the authors did not follow the current guidelines on how to report a systematic review (PRISMA checklist).

Specific Comments

Please make sure that you follow the current guidelines on how to conduct a systematic review. More information can be found here: http://prisma-statement.org/prismastatement/Checklist.aspx. This should be done for all 27 points presented in the PRISMA checklist.

L 2: You title should be change in order to identify this study as a systematic review, according to the latest PRISMA guidelines.

L 11 – 28: Please make sure your abstract is written following the PRISMA 2020 for Abstracts checklist.

L 13: RCTs should be abbreviated before using the term for first time.

L 29: Two of your keywords are already listed in your title.

L 39 – 40: Please make sure that it is clear throughout your paper that re-warm-up is suggested only during exercise in cold or neutral environments. On the other hand, the opposite (cooling techniques) is suggested during breaks while exercising in the heat. For instance: https://journals.humankinetics.com/view/journals/ijspp/aop/article-10.1123-ijspp.2020-0820/article-10.1123-ijspp.2020-0820.xml.

L 62: There is no information on the registration of your systematic reviews. The registration of a systematic review in an online registry (e.g., PROSPERO) it is required/necessary, according to the current guidelines. Please make sure your report this information.

L 67 – 70: Please consider reporting all four algorithms. Also, it is not clear why you used the term “re-warm-up” differently in the second term “halftime rewarm-up” instead of searching for just “rewarm-up”.  In addition to this, it is not clear why sometimes you searched only for the singular term (“half-time strategy”) while other ones you searched for a plural term (“RWU strategies”). An algorithm for a systematic review should be able to identify both singular and plural terms, as well as it should no be too specific.

L 72 – 78: Please consider reporting the exclusion criteria as well.

In general, I would suggest to synthesize your findings following on of the methods presented in this guidance: https://www.lancaster.ac.uk/media/lancaster-university/content-assets/documents/fhm/dhr/chir/NSsynthesisguidanceVersion1-April2006.pdf.

Reviewer 2 Report

The paper is rather an enumeration of the findings than a systematic review. The authors from 693 studies selected only 14 papers: only 2 %. This suggests, that either the first selection was too broad, or the selection criteria on the second level were too strict. The authors excluded reviews, abstracts, PhD thesis and reviews, according to the definition the systematic review identifies all relevant studies, both published and unpublished this is a strong argument pointing to the fact that this paper is not a systematic review. There is a lack of summary of the findings, clearly answering to the question: are warm-ups recommended or not? Some kind of a summary of the studies' results, for example in a table, should be given, clearly showing what sport was analysed, what type of warm-ups were used, what measures assessed, and what was the main finding. 

Author Response

REVIEWER 2

We thank this reviewer for devoting his/her time to provided us with guidance.

  1. The paper is rather an enumeration of the findings than a systematic review. The authors from 693 studies selected only 14 papers: only 2 %. This suggests, that either the first selection was too broad, or the selection criteria on the second level were too strict.

We performed a systematic search following PRISMA guidelines. Moreover, we have updated our methodology by following the 2020 guidelines, we have added a new flow chart and a new table with PICO and exclusion criteria. Finally, we have added a supplementary file summing up the search equations.

  1. The authors excluded reviews, abstracts, PhD thesis and reviews, according to the definition the systematic review identifies all relevant studies, both published and unpublished this is a strong argument pointing to the fact that this paper is not a systematic review.

It is our understanding that including abstracts, thesis and grey literature is a characteristic of a scoping review. We intended to perform a systematic review focused on RCTs, as these investigations provides high quality results. It is very frequent to found systematic reviews only focused on RCTs. We followed PRISMA guidelines in doing so. Therefore, we disagree with this observation. We should also point out that we, indeed, identified grey literature, but we did not include it in our findings.

  1. There is a lack of summary of the findings, clearly answering to the question: are warm-ups recommended or not? Some kind of a summary of the studies' results, for example in a table, should be given, clearly showing what sport was analysed, what type of warm-ups were used, what measures assessed, and what was the main finding.

We have added a new table (Table 2), showing this information. In addition, we have added implications for the coaches according to our findings in the discussion section.

Reviewer 3 Report

The article is very interesting. It aims to this review was to review analyze the results of those RCTs that provided information on the effects of re-warm-up, regardless of the sport modality investigated and the athletes' level. It is rather pleasant to read, and the methodology used seems appropriate. However, some minor (and hardly avoidable) limitations should be discussed: the rational, the argumentation and the analysis of the study. Please see my comments below:

SPECIFIC COMMENTS

Introduction. Please specify the rational of the study

Materials and Methods. Please take more care about the shape of Figures 1-3

Results. Please include p- values in the results + ES values (Table1)?

Please take more care about the shape of table 1 (Table 1. Descriptive characteristics of the studies that proposed HT interventions).

Discussion.Please  develop the discussion section. I think more elaboration on the practical implications for coaches, …

Conclusion. The conclusion is not constructive, it had to be reformulated.

Round 2

Reviewer 1 Report

Thank you for considering my suggestions and for presenting a wonderful study.

Reviewer 2 Report

The authors substantially improved their paper, especially addition of the Table 2 is an added value. In my opinion after all the changes introduced by the authors the paper can be published.